# Decreasing death rates and causes of death in Icelandic children—A longitudinal analysis

**Marina Ros Levy**[1], **Valtyr Thors**[1,2], **Sigríður Haralds Elínardottir**[3], **Alma D. Moller**[3], **Asgeir Haraldsson**[1,2] *

**1** Faculty of Medicine, University of Iceland, Reykjavík, Iceland, **2** Children's Hospital Iceland, Landspítali University Hospital Iceland, Reykjavík, Iceland, **3** Directorate of Health, Reykjavík, Iceland

* asgeir@landspitali.is

**Data Availability Statement:** All relevant data are within the paper and its Supporting Information files. (personalized data is not available). In addition, the following has been added to the

## Abstract

### Background

Global death rate in children has been declining during the last decades worldwide, especially in high income countries. This has been attributed to several factors, including improved prenatal and perinatal care, immunisations, infection management as well as progress in diagnosis and treatment of most diseases. However, there is certainly room for further progress. The aim of the current study was to describe the changes in death rates and causes of death in Iceland, a high-income country during almost half a century.

### Methods

The Causes of Death Register at The Directorate of Health was used to identify all children under the age of 18 years in Iceland that died during the study period from January 1st, 1971 until December 31st, 2018. Using Icelandic national identification numbers, individuals could be identified for further information. Hospital records, laboratory results and post-mortem diagnosis could be accessed if cause of death was unclear.

### Findings

Results showed a distinct decrease in death rates in children during the study period that was continuous over the whole period. This was established for almost all causes of death and in all age groups. This reduction was primarily attributed to a decrease in fatal accidents and fewer deaths due to infections, perinatal or congenital disease as well as malignancies, the reduction in death rates from other causes was less distinct. Childhood suicide rates remained constant.

### Interpretation

Our results are encouraging for further prevention of childhood deaths. In addition, our results emphasise the need to improve measures to detect and treat mental and behavioural disorders leading to childhood suicide.

paper: Underlying, anonymized data is accessible at https://osf.io/jy2h8/.

**Funding:** The authors received no specific funding for this work.

**Competing interests:** The authors have declared that no competing interests exist.

## Introduction

In the year 2018, 6.2 million children younger than 15 years old died worldwide, decreasing from 14.2 million in 1990 [1–3]. More than 2/3 of these children were in low and middle-income countries (LMIC), especially in sub-Saharan Africa and central and south-Asia [4]. A large part of these deaths was from preventable diseases, primarily in the youngest age groups [1].

Overall, death rates are highest during the first year of life, slowly decreasing with age but increases again between 15–18 years of age. High-income countries (HIC) generally have much lower childhood death rates [1]. In an earlier Icelandic study, the death rates and causes of death during 1941–1975 were described [5]. That study revealed a substantial reduction of childhood deaths due to infections in the first decade of the study period, i.e. during introduction of antibiotics to hospital care. Data from Iceland, ranging even further back described under five mortality rate of 360:1000 live born children in the year 1840, slowly but steadily declining to around 140:1000 children in the year 1900 [6]. The under-five mortality rate in Iceland in 2018 was 2.0:1000 and childhood death rate was 0.8:1000 live born children age 5–14 years and was comparable to the other Nordic and high income countries [2]. This data shows that death rates have fallen in Iceland over the last decades and the last century.

Causes of death in childhood differs between countries and world regions. In HIC, where many diseases in infancy can be prevented, the main causes of death are related to congenital diseases, infections and external causes such as accidents and suicide [1,7]. Important changes in death rates in children and causes of death have been recognised worldwide during the last decades [1]. In the current study we investigated the main hypothesis that an important decrease in the childhood death rates in Iceland during almost fifty years was primarily driven by fewer fatal infections in young children and childhood accidents and lower neonatal mortality.

The aim of the current study was to describe changes in childhood death rates in Iceland and evaluate causes of death over almost five decades. The underlying hypothesis was that that the declining death rates are primarily driven by decreases in fatal infections, childhood accidents, and neonatal mortality, especially in the under-five age group. From the results we attempt to determine specific risk factors and identify potential interventions to further prevent childhood deaths.

## Material and methods

In the current study, causes of death in children (from birth until their 18[th] birthday) were collected from The National Causes of Death register in Iceland. Death rates were calculated and classified according to year of death, age-groups and gender. Death rates were calculated and given as rates of death per thousand children. Information on number of living children each year and stratified to age and gender was obtained from Statistics Iceland (statice.is).

The study period was from January 1[st], 1971 until December 31[st], 2018. The National causes of death register in Iceland, kept at the Directorate of Health, Iceland is maintained by qualified personnel. All deaths are registered and coded under Icelandic national identification numbers which were used to identify all deceased individuals. Classification of diseases was done using International Classification of Diseases ICD-8 during 1971–1981, ICD-9 during 1981–1996 and ICD-10 from 1996. If cause of death was classified as "other ill-defined and unspecified causes of mortality" (ICD 10: R99) by the register, hospital records, laboratory and post-mortem results were accessed by the authors.

Causes of death were categorised according to the following groups: perinatal causes, congenital diseases, cardiovascular diseases, malignancies, infections, neurological and

degenerative diseases, other diseases, accidents, suicide and other causes [8]. Congenital malformations of the heart, nervous system and urinary tract were all grouped under congenital diseases with other congenital malformations. When a child with malignancy died, malignancy was registered as the cause of death even if a complication of the underlying disease or treatment was a strong contributing factor.

Children were categorised in age groups according to age at death, <1 year (infant), 1 to <5 years (under five), 5 to <13 years and 13 to <18 years of age.

Statistical analyses were done with R (ed 3.6.2) and R-studio (ed 1.2.5033) [9]. From The National Causes of Death register in Iceland and from population data from Statistics Iceland, death rate for each year, age groups and gender were calculated. This ratio was also calculated for the total mortality rate as well as for under five mortality rates. For analysing changes in death rates over the study period, Poisson regression (logarithmic transformed) was used with robust standard error. Two regression models were applied, one for all children 0–18 years old and another one for children under 5 years of age. The regression model for children of all ages was done with an interaction variable between genders. ANOVA was done to compare the results with and without that variable. To calculate changes in death rates and gender ratios within the groups of causes of death as well as changes within age groups, a Chi-squared test was used comparing the first decade (1971–1980) to the last one (2009–2018). Statistical significance was set at p<0.05 and a 95% confidence interval not crossing 1 for risk ratios.

The study was approved by the National Bioethics Committee (VSNb2019110034/03.01), The University Hospital's research committee and the Directorate of Health in Iceland.

## Results

The total number of children that died during the study period 1971–2018 was 2003, 1209 boys (60.4%) and 794 girls (39.6%). More than half of the children died before one year of age (1060, 53%), 57% of them were boys (599). Deceased children between one and five years of age were 285 (14% of total deaths, 58% boys), children between five and thirteen years old were 294 (15%, 63% boys) and 13 to 18 years were 364 (18%, 72% boys) (Fig 1). The death rate decreased significantly over the study period. For boys the ratio decreased from 1.5:1000 boys in 1971 to 0.18 in 2018 and for girls from 0.9 to 0.17:1000 girls in the same years (Table 1). The gender death ratio (boys:girls) was 1.5 for the whole period, declining from 1.6 in the first period to 1.1 in the last one. Boys were more likely to die during the whole period (RR: 1.45, CI: 1.33–1.59).

A significant decrease in death rate was seen for all age groups (p<0.0001) (Fig 1). The death rate was highest for under one year of age, being 5.1:1000 live born infants but declined from 10.5 in the first decade to 1.8 in the last decade.

Children under five years of age who died during the period were 1345 (67%), 57% of whom were boys (763). Under five mortality rates declined from 3.14 to 0.36:1000 over the study period, a reduction of 88.6%. These rates decreased from 3.5 to 0.4:1000 for boys and for girls from 2.8 to 0.32:1000. The gender ratio was 1.3 for the whole period, declining from 1.3 to 0.9.

The most common causes of death over the study period were perinatal causes or complications thereof (Table 1, Fig 2), accounting for 525 children (26%), almost all of whom died during the first year of life.

Over the study period, 447 children (22%) died in accidents, of which 72% were boys (320). Traffic accidents were most common, 242 (54%). The death rate caused by accidents was the highest for the age group 13-<18 years, 18.9:1000 children (Fig 2, Table 1). The gender ratio in

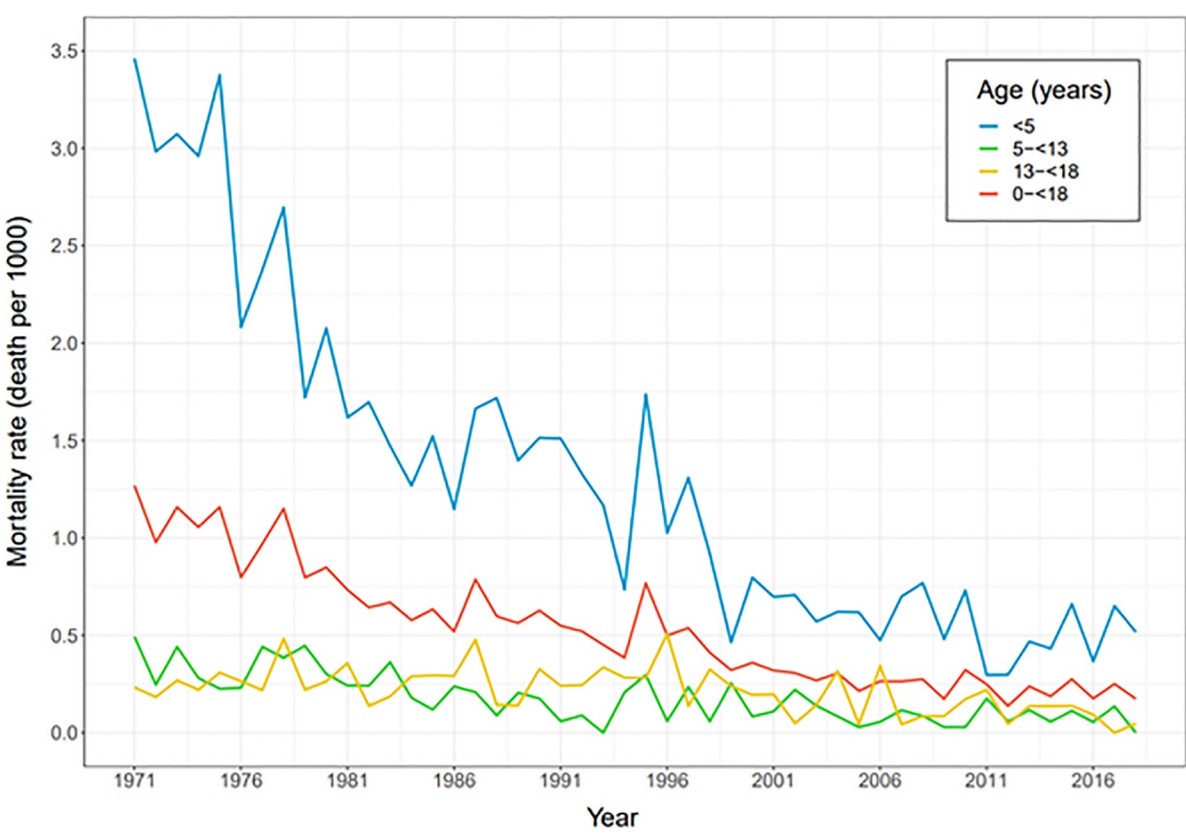

**Fig 1. Mortality rate in Icelandic children according to age groups from January 1st, 1971 until December 31st 2018.**

accidents (boys:girls) was 3.42 (p<0.0001) in the first decade but 0.96 (p = 0.91) in the last decade. The death rate from accidents declined from 0.24:1000 children (n = 188) in the first decade to 0.04 (n = 32) in the last decade (p<0.0001).

The gender ratios are seen for all causes and four different categories in Fig 3.

Congenital diseases were the cause of death in 371 children (18.5% 54% boys), 76% of them died during the first year of life (282). The death rate due to congenital diseases decreased from 0.2:1000 children in the first decade to 0.03:1000 children in the last decade (Table 1). Under five mortality rates due to congenital diseases decreased from 0.7 to 0.1:1000 children. Congenital heart diseases were the most common (39%, 144), followed by malformations of the nervous system (17%, 64) and the urinary tract (7%, 25).

When comparing the first decade to the last one, a significant decrease in death rate was observed for infections (p<0.0001) and malignancies (p<0.0001). A decrease in death rates, although not significant was found for neurological and degenerative diseases (76, 4%) and cardiovascular diseases (35, 2%) (Tables 1 and S1).

During the study period, 56 children (82% boys) committed suicide. The death rate due to suicide did not decline over the study period (p = 0.51).

In the category "Other causes" (145, 7%), sudden infant death accounted for 100 deaths (69%) in this group and avalanches caused 16 deaths (11%), of those 12 occurred in the same year in two different avalanches. Undefined causes of death were 22 (15% of this group, 1% of total) and violence caused seven deaths (5%).

Underlying, anonymised data is accessible at https://osf.io/jy2h8/.

Table 1. Causes of death in Icelandic children according to disease categories, gender and time periods from January 1st, 1971 until December 31st 2018.

| Time period* | 1971–1978 | | 1979–1988 | | 1989–1998 | | 1999–2009 | | 2009–2018 | | Total | |
|---|---|---|---|---|---|---|---|---|---|---|---|---|
| | Boys | Girls | Boys | Girls | Boys | Girls | Boys | Girls | Boys | Girls | Boys | Girls |
| | (n = 425) | (n = 248) | (n = 320) | (n = 197) | (n = 238) | (n = 172) | (n = 132) | (n = 96) | (n = 94) | (n = 81) | (n = 1209) | (n = 794) |
| **Perinatal causes** | | | | | | | | | | | | |
| Total deaths | 121 (28%) | 84 (34%) | 62 (19%) | 52 (26%) | 66 (28%) | 44 (26%) | 33 (25%) | 26 (27%) | 19 (20%) | 18 (22%) | 301 (25%) | 224 (28%) |
| Mortality rate† | 0,375 | 0,273 | 0,160 | 0,140 | 0,167 | 0,117 | 0,082 | 0,068 | 0,046 | 0,046 | 0,157 | 0,122 |
| **Accidents** | | | | | | | | | | | | |
| Traffic | 49 (42%) | 21 (60%) | 52 (50%) | 18 (62%) | 31 (53%) | 16 (62%) | 15 (58%) | 18 (86%) | 10 (62%) | 12 (75%) | 157 (49%) | 85 (67%) |
| Drowning | 30 (26%) | 2 (6%) | 20 (19%) | 4 (14%) | 14 (24%) | 5 (19%) | 4 (15%) | 1 (5%) | 1 (6%) | 2 (12.5%) | 69 (22%) | 14 (11%) |
| Other | 37 (32%) | 12 (34%) | 31 (30%) | 7 (24%) | 14 (24%) | 5 (19%) | 7 (27%) | 2 (9%) | 5 (31%) | 2 (12.5%) | 94 (29%) | 28 (22%) |
| Total deaths | 116 (27%) | 35 (14%) | 103 (32%) | 29 (15%) | 59 (25%) | 26 (15%) | 26 (20%) | 21 (22%) | 16 (17%) | 16 (20%) | 320 (26%) | 127 (16%) |
| Mortality rate† | 0,359 | 0,114 | 0,265 | 0,078 | 0,149 | 0,069 | 0,065 | 0,055 | 0,039 | 0,041 | 0,167 | 0,069 |
| **Congenital diseases** | | | | | | | | | | | | |
| Circulatory system | 26 (40%) | 21 (37%) | 28 (39%) | 21 (40%) | 17 (45%) | 14 (40%) | 3 (21%) | 5 (42%) | 4 (36%) | 5 (33%) | 78 (39%) | 66 (39%) |
| Nervous system | 12 (18%) | 13 (23%) | 14 (19%) | 8 (15%) | 2 (5%) | 4 (11%) | 2 (14%) | 2 (17%) | 1 (9%) | 6 (40%) | 31 (15.5%) | 33 (19%) |
| Other | 27 (42%) | 23 (40%) | 30 (42%) | 23 (44%) | 19 (50%) | 17 (49%) | 9 (64%) | 5 (42%) | 6 (55%) | 4 (27%) | 91 (45.5%) | 72 (42%) |
| Total deaths | 65 (15%) | 57 (23%) | 72 (22%) | 52 (26%) | 38 (16%) | 35 (20%) | 14 (11%) | 12 (12%) | 11 (12%) | 15 (19%) | 200 (17%) | 171 (22%) |
| Mortality rate† | 0,201 | 0,185 | 0,185 | 0,140 | 0,096 | 0,093 | 0,035 | 0,031 | 0,027 | 0,038 | 0,104 | 0,093 |
| **Infections** | | | | | | | | | | | | |
| Respiratory | 23 (45%) | 12 (40%) | 4 (27%) | 1 (6%) | 5 (45%) | 4 (33%) | 1 (17%) | 3 (38%) | 1 (20%) | 1 (33%) | 34 (39%) | 21 (30%) |
| Sepsis | 15 (29%) | 8 (27%) | 2 (13%) | 2 (12%) | 2 (18%) | 4 (33%) | 0 (0%) | 3 (38%) | 3 (60%) | 1 (33%) | 22 (25%) | 18 (26%) |
| Other | 13 (25%) | 10 (33%) | 9 (60%) | 13 (81%) | 4 (36%) | 4 (33%) | 5 (83%) | 2 (25%) | 1 (20%) | 1 (33%) | 32 (36%) | 30 (43%) |
| Total deaths | 51 (12%) | 30 (12%) | 15 (5%) | 16 (8%) | 11 (5%) | 12 (7%) | 6 (5%) | 8 (8%) | 5 (5%) | 3 (4%) | 88 (7%) | 69 (9%) |
| Mortality rate† | 0,158 | 0,098 | 0,039 | 0,043 | 0,028 | 0,032 | 0,015 | 0,021 | 0,012 | 0,008 | 0,046 | 0,038 |
| **Malignancies** | | | | | | | | | | | | |
| Central nervous system | 5 (21%) | 7 (41%) | 3 (17%) | 4 (36%) | 2 (33%) | 4 (44%) | 5 (33%) | 4 (44%) | 4 (44%) | 2 (50%) | 20 (28%) | 21 (42%) |
| Leukemia | 9 (38%) | 5 (29%) | 9 (50%) | 3 (27%) | 3 (50%) | 2 (22%) | 2 (13%) | 3 (33%) | 2 (22%) | 2 (50%) | 25 (35%) | 15 (30%) |
| Other | 10 (42%) | 5 (29%) | 6 (33%) | 4 (36%) | 1 (17%) | 3 (33%) | 8 (53%) | 2 (22%) | 3 (33%) | 0 (0%) | 27 (38%) | 14 (28%) |
| Total deaths | 24 (6%) | 17 (7%) | 18 (6%) | 11 (6%) | 6 (3%) | 9 (5%) | 15 (11%) | 9 (9%) | 9 (10%) | 4 (5%) | 72 (6%) | 50 (6%) |
| Mortality rate† | 0,074 | 0,055 | 0,046 | 0,030 | 0,015 | 0,024 | 0,037 | 0,023 | 0,022 | 0,010 | 0,038 | 0,027 |
| **Neurological and degenerative diseases** | | | | | | | | | | | | |
| Total deaths | 10 (2%) | 8 (3%) | 8 (2%) | 4 (2%) | 12 (5%) | 5 (3%) | 9 (7%) | 2 (2%) | 9 (10%) | 9 (11%) | 48 (4%) | 28 (4%) |
| Mortality rate† | 0,031 | 0,026 | 0,021 | 0,011 | 0,030 | 0,013 | 0,022 | 0,005 | 0,022 | 0,023 | 0,025 | 0,015 |
| **Other diseases** | | | | | | | | | | | | |
| Total deaths | 14 (3%) | 11 (4%) | 5 (2%) | 6 (3%) | 7 (3%) | 6 (3%) | 4 (3%) | 7 (7%) | 5 (5%) | 4 (5%) | 35 (3%) | 34 (4%) |
| Mortality rate† | 0,043 | 0,036 | 0,013 | 0,016 | 0,018 | 0,016 | 0,010 | 0,018 | 0,012 | 0,010 | 0,018 | 0,019 |
| **Suicide** | | | | | | | | | | | | |
| Total deaths | 6 (1%) | 1 (0%) | 11 (3%) | 0 (0%) | 14 (6%) | 6 (3%) | 6 (5%) | 1 (1%) | 9 (10%) | 2 (2%) | 46 (4%) | 10 (1%) |
| Mortality rate† | 0,019 | 0,003 | 0,028 | 0,000 | 0,035 | 0,016 | 0,015 | 0,003 | 0,022 | 0,005 | 0,024 | 0,005 |
| **Cardiovascular diseases** | | | | | | | | | | | | |
| Total deaths | 6 (1%) | 2 (1%) | 4 (1%) | 5 (3%) | 1 (0%) | 4 (2%) | 5 (4%) | 2 (2%) | 5 (5%) | 1 (1%) | 21 (2%) | 14 (2%) |
| Mortality rate† | 0,019 | 0,007 | 0,010 | 0,013 | 0,003 | 0,011 | 0,012 | 0,005 | 0,012 | 0,003 | 0,011 | 0,008 |
| **Other causes** | | | | | | | | | | | | |

(Continued)

**Table 1.** (Continued)

| Time period* | 1971–1978 | | 1979–1988 | | 1989–1998 | | 1999–2009 | | 2009–2018 | | Total | |
|---|---|---|---|---|---|---|---|---|---|---|---|---|
| | Boys | Girls | Boys | Girls | Boys | Girls | Boys | Girls | Boys | Girls | Boys | Girls |
| | (n = 425) | (n = 248) | (n = 320) | (n = 197) | (n = 238) | (n = 172) | (n = 132) | (n = 96) | (n = 94) | (n = 81) | (n = 1209) | (n = 794) |
| Total deaths | 12 (3%) | 3 (1%) | 22 (7%) | 22 (11%) | 24 (10%) | 25 (15%) | 14 (11%) | 8 (8%) | 6 (6%) | 9 (11%) | 78 (6%) | 67 (8%) |
| Mortality rate† | 0,037 | 0,010 | 0,057 | 0,059 | 0,061 | 0,066 | 0,035 | 0,021 | 0,015 | 0,023 | 0,041 | 0,037 |
| **Total deaths** | **425 (100%)** | **248 (100%)** | **320 (100%)** | **197 (100%)** | **238 (100%)** | **172 (100%)** | **132 (100%)** | **96 (100%)** | **94 (100%)** | **81 (100%)** | **1209 (100%)** | **794 (100%)** |
| **Mortality rate†** | **1,316** | **0,806** | **0,824** | **0,532** | **0,601** | **0,456** | **0,329** | **0,249** | **0,230** | **0,207** | **0,630** | **0,433** |

*From 1.jan 1971–31. dec 2018. †Per 1000 live births (0–18 years).

## Discussion

The main finding of this study was a striking decrease in death rates among children up to 18 years of age during the study period of almost fifty years. This was established for almost all

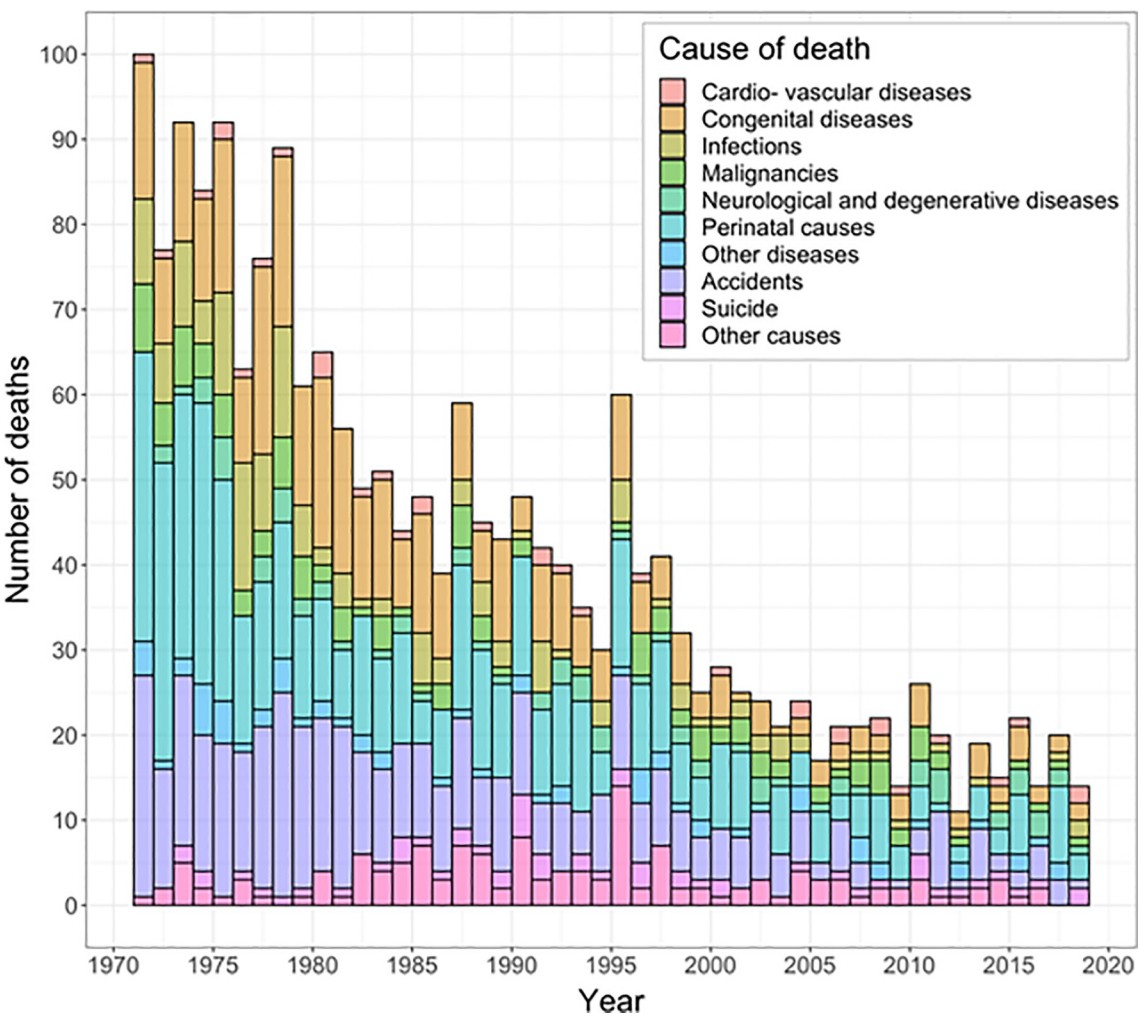

**Fig 2. Number of deaths in Icelandic children according to disease categories from January 1st, 1971 until December 31st 2018.**

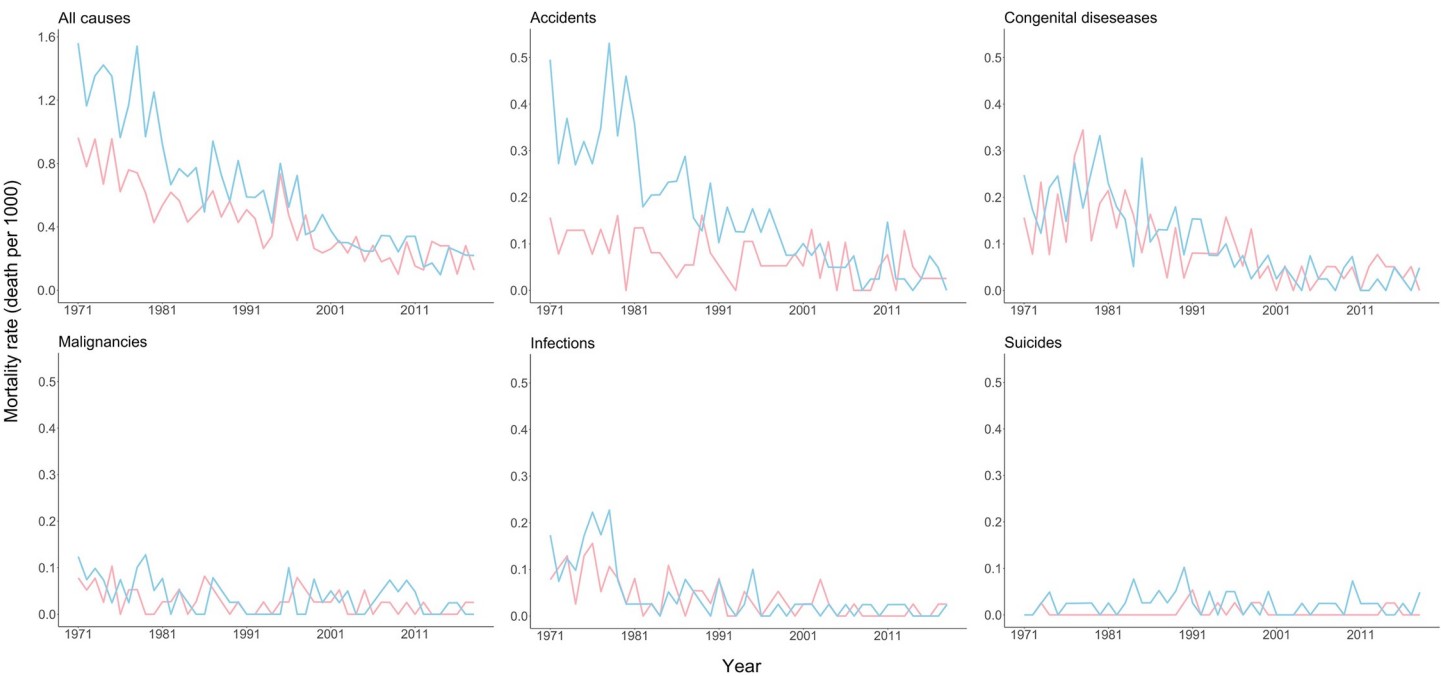

**Fig 3. Gender ratios for all causes of childhood deaths and four different categories in Iceland during 1971–2018.**

causes of death and in all age groups. The trend appears to be consistently negative over the period.

The gender ratio clearly revealed higher risk of death for boys than girls, most obvious in the adolescent age group and more pronounced in the first decades of the study period. These ratios have been described by others [10,11]. This gender difference is less clear in under five mortality rates as has also been described by UNICEF in HIC [1]. In our study, a decline in male/female ratio was seen from 1.6 to 1.2 and deserves special attention. The reason may be due to behavioural differences as well as physiological, social and environmental factors. This is reflected in the higher death rate for boys due to accidents. The changes in gender ratios were clearly reflected in the decreasing death ratio caused by accidents among boys. These causes have been identified in several countries [8]. During the study period, several safety measures for accident prevention were implemented, especially regarding traffic safety. These safety measures as well as increased awareness are probably the main factors leading to the decreased childhood mortality due to accidents.

Under five mortality rates have been declining, especially in HIC according to WHO and UNICEF [1]. This is in accordance with our findings. In a recent report from WHO [1], the decrease in under five mortality rates in Europe and USA in 1990–2018 was 72% and 42% respectively. In our study this decline was 67% during the same period. Several factors contribute to this decline; improved prenatal health care and perinatal survival, infection control and progress in health care in general being the most obvious reasons [1,11]. Perinatal and neonatal deaths in Iceland have been substantially decreasing over the period and is currently among the lowest in Europe [12]. This obviously contributes to the improved survival in this age group.

The highest risk of mortality was in the age group less than one year of age followed by the oldest age group, 13 to <18 years. This is comparable to other studies in HIC [13,14]. The reason for the relative high death rate in the youngest group may be attributed to perinatal and

neonatal events as well as congenital disease. In HIC, premature births have become more common, but the survival rate of premature babies has improved. This has also been found in Iceland [12]. Nevertheless, perinatal events may still cause some of these early deaths [2,14,15]. The causes of death in the oldest age group are most often related to long term illnesses, accidents or suicide.

Accidents were the second most common cause of death over the study period. An impressive decrease in deaths due to accidents, especially traffic accidents, was noted. The gender difference was particularly obvious in the first decade of the study but was almost equal in the final decade. Similar findings have been described in other Nordic and European countries [16,17]. Traffic accidents were the most common cause of accidents and accounted for more than half of the fatal accidents. Similar results for the youngest age groups have been reported from other Nordic countries [8]. The fact that boys are more prone to fatal accidents probably reflect gender differences in risk taking behaviour [7,10,11,18]. Although the number of fatal accidents in children are still too high, the rate has declined significantly. The reason for this is presumably higher risk awareness, better training and education of young drivers, improved traffic and car safety measures. Nevertheless, this still leaves room for improvement.

In our study, sixteen children lost their lives in avalanches. These children were classified separately as natural disaster rather than accidents as they are different in nature and harder to prevent or predict.

Congenital diseases make up for the third most common cause of death in our study, almost all occurring before the age of five and most were cardiac malformations. However, there was a substantial decline during the study period. This is in accordance with other findings, especially in countries where prenatal diagnosis of severe diseases is applied with a possibility of preparing for delivery of a baby with severe congenital malformations or termination of the pregnancy [19–22].

Death caused by infections declined significantly during the study period. This can be attributed to better diagnosis and improved antibiotic treatment as well as introduction of new vaccines in universal immunisations schedules. An earlier Icelandic study, describing death rate and causes of death during 1941–1975, revealed a substantial reduction of fatal infections in the first decade of that study, i.e. during increasingly widespread use of antibiotics [5]. The last measles epidemic in Iceland was in 1977 and immunisation coverage in Iceland is very high [23,24]. This has obviously contributed to declining incidence of fatal infections in Icelandic children [25–27]. This clearly underlines the success of immunisations and appropriate antibiotic usage.

A decline in deaths due to all categories of childhood malignancies was confirmed in our study. The Nordic Society of Pediatric Haematology and Oncology (NOPHO) reports incidence and treatment results of childhood malignancies in the Nordic countries [28]. The results in our study are in context with the findings from NOPHO and other HIC [8,28–30]. As the incidence of malignancies in children has not decreased, this success must be attributed to improved cancer treatment as well as better supportive care, including infection management.

The decline in deaths due to cardiovascular diseases and neurological disorders was not statistically significant. However, these are small numbers and results must be interpreted with caution.

The rate of children dying due to suicide did not change significantly during the study period. Similar findings have been reported by others [31,32] although other Nordic countries have reported a decreasing incidence [33]. As in other studies, boys are at much higher risk than girls [31–33]. This finding is of a grave concern. We recognise that determining whether a cause of death should be categorised as suicide is often difficult. The available data may not

always be sufficient to distinguish between accidents (e.g. by drug overdose) and suicides. Suicide may therefore possibly be underestimated, especially in teenagers. However, heightened awareness of suicidal risk in teenagers is important for implementing preventive measures. This finding of no or slim reductions in death rates due to suicide in our study must be a strong encouragement to increase our awareness of the importance of mental health and psychological disorders in children in order to prevent these tragic events. A working group on suicide preventions issued by the Icelandic government released a program of action in 2018 with one of the goals being to ensure regular and evidence-based education on mental health in day-care centres, as well as primary and secondary schools. This, among other measures, will hopefully lead to better awareness and bring us one step closer to intervene and provide help to those who need it before it is too late.

A major strength of this study is the utilisation of nationwide and complete data source of causes of death over an almost fifty-year period. The Causes of Death register is kept at the Directorate of Health in Iceland. The register contains individual-level information from all death certificates and from autopsy reports if any autopsy was performed. Every death is coded manually by one specially trained personnel. Cases of uncertain cause of death were reviewed by a specially trained physician. In order to support the categorisation of underlying cause of death the Acme user interface programme is used to check manual coding. In addition, using the national identification numbers, we were able to find hospital records, laboratory results or post-mortem diagnosis in cases where the death certificate was unclear. We therefore maintain that the results are comprehensive and reflect nationwide childhood fatality rates with minimal errors of classifications.

Although we are confident of the accuracy of the data and the total number of cases reviewed exceeded two thousand, some of the categories are rather small. This is inevitable in a small country as Iceland. Nevertheless, our findings are in concordance with other published studies.

The results of this study show a clear reduction in death rates in children over an almost fifty-year period. This reduction is primarily attributed to a decrease in fatal accidents, but also to fewer deaths due to infections, perinatal or congenital disease and malignancies although a reduction in other causes was also noted. Our results should be encouraging to improve this outcome even further but also to increase emphasis on further preventing childhood mortality, including measures to detect mental and behavioural disorders which may lead to childhood suicide.

## Supporting information

**S1 Table. Causes of death in Icelandic children according to disease categories and sub-categories, gender and time periods from January 1st, 1971 until December 31st 2018.** (DOCX)

## Acknowledgments

Benedikt Th Sigurjonsson at The University of Iceland contributed to the statistical analysis.

## Author Contributions

**Conceptualization:** Valtyr Thors, Sigríður Haralds Elínardottir, Alma D. Moller, Asgeir Haraldsson.

**Data curation:** Marina Ros Levy, Valtyr Thors, Sigríður Haralds Elínardottir, Asgeir Haraldsson.

**Formal analysis:** Marina Ros Levy, Valtyr Thors, Asgeir Haraldsson.

**Investigation:** Valtyr Thors, Asgeir Haraldsson.

**Methodology:** Valtyr Thors, Sigríður Haralds Elínardottir, Alma D. Moller, Asgeir Haraldsson.

**Project administration:** Asgeir Haraldsson.

**Resources:** Marina Ros Levy.

**Supervision:** Valtyr Thors, Asgeir Haraldsson.

**Validation:** Marina Ros Levy, Valtyr Thors, Sigríður Haralds Elínardottir, Alma D. Moller, Asgeir Haraldsson.

**Writing – original draft:** Marina Ros Levy, Valtyr Thors, Sigríður Haralds Elínardottir, Alma D. Moller, Asgeir Haraldsson.

**Writing – review & editing:** Marina Ros Levy, Valtyr Thors, Sigríður Haralds Elínardottir, Alma D. Moller, Asgeir Haraldsson.

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
