## [Decision Letter · Decision Letter 0]

11 Mar 2021

PONE-D-20-37583

Decreasing death rates and causes of death in Icelandic children - A longitudinal analysis

PLOS ONE

Dear Dr. Haraldsson,

Thank you for submitting your manuscript to PLOS ONE. After careful consideration, we feel that it has merit but does not fully meet PLOS ONE’s publication criteria as it currently stands. Therefore, we invite you to submit a revised version of the manuscript that addresses the points raised during the review process.

Specifically :

Please elaborate your answer to all of the methodology section comments.Please make sure the data underlying the findings in this manuscript are fully available.

We look forward to receiving your revised manuscript.

Kind regards,

Amir Radfar, MD,MPH,MSc,DHSc

Academic Editor

PLOS ONE

Journal Requirements:

3. Please include a copy of Table II which you refer to in your text on page 5.

Reviewers' comments:

Reviewer's Responses to Questions

**Comments to the Author**

1. Is the manuscript technically sound, and do the data support the conclusions?

Reviewer #1: Partly

Reviewer #2: Yes

Reviewer #3: Yes

2. Has the statistical analysis been performed appropriately and rigorously? 

Reviewer #1: I Don't Know

Reviewer #2: Yes

Reviewer #3: No

3. Have the authors made all data underlying the findings in their manuscript fully available?

Reviewer #1: No

Reviewer #2: No

Reviewer #3: No

4. Is the manuscript presented in an intelligible fashion and written in standard English?

Reviewer #1: Yes

Reviewer #2: Yes

Reviewer #3: Yes

5. Review Comments to the Author

Reviewer #1: In the manuscript “Decreasing death rates and causes of death in Icelandic children—A longitudinal analysis”, the authors analyze a near 50 year dataset of mortality data to ascertain how death rates have changed over the study period. More specifically, the authors subset the data into sex and age groups, calculate mortality rates for various causes of death, and compare how and/or model how how rates have changed over the period. This is an interesting topic and use of public health data, as it has the potential to evaluate the affect of past public health policies. However, several major aspects of the current paper prevents me from recommending it for publication in this journal.

First, the trove of data the authors use is certainly worth analysis, and Figures 1 and 2 are interesting in and of themselves. However, the authors do not establish any sort of questions or hypotheses that they wish to answer or evaluate by doing the analysis. Thus, it feels more like a preliminary data exploration exercise than a mature, hypothesis driven analysis. There is an implicit hypothesis, captured in one of the statistical methods used (Chi-square), that the mortality rates at the end of the study period differ from those at the beginning. However, the lack of explicit questions or hypotheses results in a disappointingly short exploration of some of the more surprising results, such as the marginal increase in death rates due to suicide in recent decades. I would recommend re-examining the whole of the data, and more thoroughly considering which patterns are expected versus ones that are not, and thus suggest the need for re-evaluation of public health policy.

Second, the statistical methods chosen seem reasonable given the data, but how the data were processed, how rates were calculated, and how the statistical methods were applied to the data are very much unclear. I was also frequently uncertain whether the presented results pertained to the Chi-square or the Poisson regression analyses. Further, some of the results discussed in the text did not seem to align with the data in the tables. For example, in paragraph 1 of the results, it says “The death rate decreased significantly over the study period, for boys the ratio decreased from 1·5:1000 boys to 0·18 and for girls from 0·9 to 0·17:1000 girls Table I)”. However, the table shows different values for boys and girls in the first and last decade examined. I suppose this sentence was referring to the first and last years, but this isn't clear. And, the figure does not differentiate between girls and boys, so that doesn't help clarify things. To evaluate the rigor of a study, it is imperative that statistical methodologies are clearly explained, and that there is an unambiguous connection between analyses conducted and results presented.

Next, the results are presented in a “laundry-list” sort of manner that may be more appropriate for a journal of public health statistics. In addition, organization of the results section into subsections would significantly improve the readability of that section.

Lastly, the discussion section does discuss results in the context of other HIC, and some reasonable interpretations for how past interventions could have resulted in the patterns shown are made. However, it does not “determine specific risk factors and analyse possible interventions” as suggested in the introduction. The determination of risk factors and determining important contributors to mortality rates could be done via appropriate multivariate modeling. This is not done here, as I far as I can tell. In addition, the discussion of interventions for some causes of mortality, such as from accidents, are discussed only vaguely and lack citations.

Minor comments: there are no line numbers on the PDF of the submission. This makes it difficult to refer to specific sections. Therefore, I will refer to paragraphs of sections.

Abstract: The word “impressive” in inappropriate here, as it has an emotional connotation.

Findings section: what does “a less impressive reductions was seen in other causes” mean?

Introduction: the last paragraph is a single sentence. It should be split into several sentences for clarity.

Methods:

1st paragraph: causes of death weren't described in this study, they were compiled from records.

2nd paragraph: What is the ICD-8 through 10 system? Is it a relevant detail here?

1. In the case of “ill-defined and unspecified causes of mortality”, how were hospital records used to assign cause of death? Was this done in a consistent way by a qualified person? Is there uncertainty in this process, and what proportion of the data is made up of this classification

3rd paragraph:Were these categories determined using a previously published methodology, or was it done in an adhoc way?

5th paragraph: Since the entire study is about mortality rates, I would suggest explicitly writing how all rates were calculated(year, decade, group, sex, age), and which rates were used in what analysis.

Results:

1st paragraph: “Children aged 1-<5 years were 285 (14%, 58% boys), aged 5-<13 years old were 294 (15%, 63% boys) and 13-<18 years were 364 (18%, 72% boys) (Figure 1).” Is this referring to children who died?

2nd paragraph: “Boys were 41% more likely to die during the whole period (CI: 0·51-0·67).” How was this calculation made?

5th paragraph: This should be split into several paragraphs with different sections to guide the reader.

Overall: I would suggest additional tables and/or figures to help cross walk the age and gender information. As it is, you cannot tell at what age boys and girls diverge in terms of suicide-based mortality. But this is important to a prevention measures.

Figure 1: Why are there only three lines when 4 age classes are considered. Also, the age classes shown in the legend are the not the same as in the text.

Supplemental Table: This table appears to contain the same information as the Table 1 along with addition information. Why not just have 1 table?

Discussion:

2nd paragraph: “The gender ratio revealed clearly a higher risk for boys than girls, most obvious in the oldest age group and more pronounced in earlier periods”. This seems contradictory.

4th paragraph: “In HIC, the number of prenatal births has increased”; do you mean premature births?

Overall: there are several assertions made without citations or data. For example “The reason for this gender difference may be due to behavioural differences as well as physiological, social and environmental factors. The decreasing gender ratio is primarily due to less fatal accidents among boys, probably through increased awareness and accident prevention.” While speculations about mechanisms of sex-based mortality differences or intervention success have to be made here, they should be made based on literature-based sources.

Reviewer #2: This is an important study on the changes in death rates in children in Iceland. My comments are as follows

1. The manuscript is written in standard English but I do feel that some sentences are long or a little bit unclear and clarity could be improved throughout by a thorough edit.

2. Methods: Could 'cardio- vascular' just be 'cardiovascular'?

3. Methods: Could the authors provide a reference or further justification for why they grouped causes of death as they did?

4. Results: Could the authors be consistent when they provide number and when they provide per cent? Sometimes they provide both and sometimes they just provide per cent. For example, in the first sentence, both numbers and percentages are provide, while in the second, only percentages are provided.

5. Results: Paragraph 3. Sorry I may be misreading this, but I would have assumed the 95% CIs would be above 1 if boys were 41% more likely to die (1.59, 95% CI 1.51-1.67)? Could the authors put the RR with the CIs for clarity?

6. Results: I would prefer to see a p-value, rather than n.s. when the ratio is not significant.

7. Results/Discussion: Why are avalanches not grouped with accidents?

8. Discussion: In the second paragraph, the authors say the gender ratio declined from 1.8 to 1.2. In the results, it says it declined from 1.6 to 1.1. Which is correct?

9. Discussion: Second paragraph. The last sentence "This deserves special attention", leaves me asking 'why?' I would recommend the authors either delete or add more detail on why.

10. Discussion: Fourth paragraph. I got a little lost here. Weren't perinatal causes the most common cause of death? Could the authors clarify?

Reviewer #3: This manuscript presents interesting analyses of changes in death rates over time in Iceland. These analyses contain some valuable contributions, though I would suggest some changes to the analyses to make them more robust.

I also feel as though this manuscript could take the analyses further in the examples presented. Maybe this is just the epidemiologist in me, but, for instance, could the measles vaccination rates be included as a covariate in a model to determine if the rates are correlated with the declines in measles infections. For the decrease in accidents, maybe there were specific government campaigns to improving driving education or maybe just safer cars were released.

1. For the Poisson distribution, the mean and variance are the same parameter, i.e., the mean equals the variance. That can be a very poor assumption in regression models and I would strongly urge you to try a quasipossion or negative binomial model. Those distributions decouple the mean and variance and allow for the variance to be estimated independent of the mean.

2. My understanding is that you've modeled the rates. I would suggest trying to model the death counts instead of the rates with the population as an offset. This is a more natural way to model this, and you can obtain the death rates from the model and convert them to deaths per 1000.

3. I also wasn't sure why a Chi-squared test was used especially after you've been working with a model that is testing trends over time. You could utilize the model to estimate the changes from the start of the interval to the end, probably with contrast statements.

4. In the results, I would be a little more explicit about some of the sample sizes presented. The sentence about deaths from accidents is great, but it's hard to tell what the time frame is for the other diseases.

6. PLOS authors have the option to publish the peer review history of their article (what does this mean?). If published, this will include your full peer review and any attached files.

Reviewer #1: No

Reviewer #2: No

Reviewer #3: No

---

## [Author Response · Author response to Decision Letter 0]

1 Jun 2021

All comments by the reviewers have been addressed.

Underlying data has been made available online as stated in the manuscript

Table II has been removed.

---

## [Decision Letter · Decision Letter 1]

23 Jun 2021

PONE-D-20-37583R1

Decreasing death rates and causes of death in Icelandic children - A longitudinal analysis

PLOS ONE

Dear Dr. Haraldsson,

Thank you for submitting your manuscript to PLOS ONE. After careful consideration, we feel that it has merit but does not fully meet PLOS ONE’s publication criteria as it currently stands. Therefore, we invite you to submit a revised version of the manuscript that addresses the points raised during the review process.

Specifically ,please address all comments made by reviewer #1

We look forward to receiving your revised manuscript.

Kind regards,

Amir Radfar, MD,MPH,MSc,DHSc

Academic Editor

PLOS ONE

Reviewers' comments:

Reviewer's Responses to Questions

**Comments to the Author**

1. If the authors have adequately addressed your comments raised in a previous round of review and you feel that this manuscript is now acceptable for publication, you may indicate that here to bypass the “Comments to the Author” section, enter your conflict of interest statement in the “Confidential to Editor” section, and submit your "Accept" recommendation.

Reviewer #1: (No Response)

Reviewer #3: All comments have been addressed

2. Is the manuscript technically sound, and do the data support the conclusions?

Reviewer #1: Yes

Reviewer #3: (No Response)

3. Has the statistical analysis been performed appropriately and rigorously? 

Reviewer #1: I Don't Know

Reviewer #3: (No Response)

4. Have the authors made all data underlying the findings in their manuscript fully available?

Reviewer #1: Yes

Reviewer #3: (No Response)

5. Is the manuscript presented in an intelligible fashion and written in standard English?

Reviewer #1: Yes

Reviewer #3: (No Response)

6. Review Comments to the Author

Reviewer #1: The authors have reasonably responded to most of my comments and have significantly improved their manuscript. However, there are still some import issues that need to be addressed before I can recommend this manuscript for publication.

INTRODUCTION

The end of the second paragraph and the last paragraphs are a bit confusing. The author’s state that the main hypothesis that childhood death rates are decreasing due to fewer fatal infections. But then the author’s say that the underlying hypothesis is that death rates are declining. I think that what the authors are aiming to say is that the data show that death rates have fallen over the previous 5 decades and it is a hypothesis is that these declines are primarily driven by decreases in fatal infections, childhood accidents, and neonatal mortality. If this is correct, this should be clarified.

METHODS

In the statistical analysis section, the authors say they have stated how rates were calculated, but they only say in the methods that “death rate for each year, age groups and gender were calculated”. In the results, it can be seen that rates are listed per 1000 children. However, providing a simple equation for rate calculation, and/or stating that they are rates of death per thousand children in the methods section would be helpful to clear up ambiguity.

RESULTS

There is still uncertainty about the Poisson regression model that was applied to the data. Was a separate regression model fit for each group, or was a single model used with age and gender groups as factors? Which factors were significant and which factors weren’t? If multiple models were applied, was there a correction for multiple p-values. These issues could be addressed by simply reporting the terms of the model used in the text. In addition, PLoSOne guidelines for regression analyses suggest that they should be included, at least as part of the supplementary data:

Regression analyses. Include the full results of any regression analysis performed as a supplementary file. Include all estimated regression coefficients, their standard error, p-values, and confidence intervals, as well as the measures of goodness of fit.

On the same topic, providing the details of all the Chi-square analyses performed would also be helpful to fully appreciate the scope of the analysis that the authors conducted.

DISCUSSION

The statement that “the decrease was continuous over the period” does not appear to be true. There is clear variation from year to year with increases in some groups occurring several years in a row (e.g. Figure 1, group 13-18, 1982-1987). However, the trend appears to be consistently negative over the period, which is what the Poisson regression (I think?) showed.

Did the authors quantify that “The decreasing gender ratio was in fact primarily driven by fewer fatal accidents among boys”? While it is mentioned in the results that there was a dramatic decrease in the boys:girls death ratio of accidents, and figure 3 is provided to show this, I don’t see any sort of quantification of the contribution of accidents, relative to other causes of death, to differences in boy:girl death ratios between the two decades of comparison. In addition, figure 3 is at a very low resolution, making the y-axis largely unreadable. Although I can see that accidents are the only cause of death for which male and female death rates (maybe counts?) do not obviously overlap, the y-axes are clearly on different scales. A quantification here would make the author’s argument on this point stronger.

In the sentence “The reason for the relative high death rate in the youngest group may be contributed to perinatal and neonatal events as well as congenital disease”, I think the authors might want to use the word “attributed” instead of “contributed”.

For the statement “Traffic accidents were the most common cause of accidents and accounted for more than half of the fatal accidents. In the younger age groups, most victims were pedestrians, but older children were more often car passengers”, I don’t see where this is shown, as the supplementary table is not stratified by age. Since there are no limits on the size of supplementary files, it seems reasonable to include this information if it is reported in the text. Also, is the implication here that older boys would have higher traffic mortality as the drivers of cars than girls? If so, that should be stated.

In the sentence “The available data may not always be sufficient to distinguish between accidents e.g. by drug overdose and suicides”, I believe that the authors need to add a parentheses: “…accidents (e.g., by drug overdose) and suicides”.

Reviewer #3: (No Response)

7. PLOS authors have the option to publish the peer review history of their article (what does this mean?). If published, this will include your full peer review and any attached files.

Reviewer #1: No

Reviewer #3: No

---

## [Author Response · Author response to Decision Letter 1]

21 Aug 2021

All questions have been addressed in the file Decreasing death rates in children - Resp to reviewers II

---

## [Decision Letter · Decision Letter 2]

7 Sep 2021

Decreasing death rates and causes of death in Icelandic children - A longitudinal analysis

PONE-D-20-37583R2

Dear Dr. Haraldsson,

We’re pleased to inform you that your manuscript has been judged scientifically suitable for publication and will be formally accepted for publication once it meets all outstanding technical requirements.

Kind regards,

Amir Radfar, MD,MPH,MSc,DHSc

Academic Editor

PLOS ONE

Additional Editor Comments (optional):

Reviewers' comments:

Reviewer's Responses to Questions

**Comments to the Author**

1. If the authors have adequately addressed your comments raised in a previous round of review and you feel that this manuscript is now acceptable for publication, you may indicate that here to bypass the “Comments to the Author” section, enter your conflict of interest statement in the “Confidential to Editor” section, and submit your "Accept" recommendation.

Reviewer #1: All comments have been addressed

Reviewer #3: All comments have been addressed

2. Is the manuscript technically sound, and do the data support the conclusions?

Reviewer #1: Yes

Reviewer #3: (No Response)

3. Has the statistical analysis been performed appropriately and rigorously? 

Reviewer #1: Yes

Reviewer #3: (No Response)

4. Have the authors made all data underlying the findings in their manuscript fully available?

Reviewer #1: Yes

Reviewer #3: (No Response)

5. Is the manuscript presented in an intelligible fashion and written in standard English?

Reviewer #1: Yes

Reviewer #3: (No Response)

6. Review Comments to the Author

Reviewer #1: (No Response)

Reviewer #3: (No Response)

7. PLOS authors have the option to publish the peer review history of their article (what does this mean?). If published, this will include your full peer review and any attached files.

Reviewer #1: No

Reviewer #3: No

---

## [Editor Report · Acceptance letter]

22 Sep 2021

PONE-D-20-37583R2 

Decreasing death rates and causes of death in Icelandic children - A longitudinal analysis 

Dear Dr. Haraldsson:

I'm pleased to inform you that your manuscript has been deemed suitable for publication in PLOS ONE. Congratulations! Your manuscript is now with our production department. 

Kind regards, 

on behalf of

Dr. Amir Radfar 

Academic Editor

PLOS ONE